

# LOANet: a lightweight network using object attention for extracting buildings and roads from UAV aerial remote sensing images

Xiaoxiang Han[1,2,*], Yiman Liu[3,4,*], Gang Liu[5], Yuanjie Lin[2] and Qiaohong Liu[1]

[1] School of Medical Instruments, Shanghai University of Medicine and Health Sciences, Shanghai, People's Republic of China

[2] School of Health Sciences and Engineering, University of Shanghai for Science and Technology, Shanghai, People's Republic of China

[3] Department of Pediatric Cardiology, Shanghai Children's Medical Center, School of Medicine, Shanghai Jiao Tong University, Shanghai, People's Republic of China

[4] Shanghai Key Laboratory of Multidimensional Information Processing, School of Communication & Electronic Engineering, East China Normal University, Shanghai, People's Republic of China

[5] Key Laboratory of Earthquake Geodesy, Institute of Seismology, China Earthquake Administration, Wuhan, Hubei, People's Republic of China

[*] These authors contributed equally to this work.

Corresponding author
Qiaohong Liu, hqllqh@163.com

## ABSTRACT

Semantic segmentation for extracting buildings and roads from uncrewed aerial vehicle (UAV) remote sensing images by deep learning becomes a more efficient and convenient method than traditional manual segmentation in surveying and mapping fields. In order to make the model lightweight and improve the model accuracy, a lightweight network using object attention (LOANet) for buildings and roads from UAV aerial remote sensing images is proposed. The proposed network adopts an encoder-decoder architecture in which a lightweight densely connected network (LDCNet) is developed as the encoder. In the decoder part, the dual multi-scale context modules which consist of the atrous spatial pyramid pooling module (ASPP) and the object attention module (OAM) are designed to capture more context information from feature maps of UAV remote sensing images. Between ASPP and OAM, a feature pyramid network (FPN) module is used to fuse multi-scale features extracted from ASPP. A private dataset of remote sensing images taken by UAV which contains 2431 training sets, 945 validation sets, and 475 test sets is constructed. The proposed basic model performs well on this dataset, with only 1.4M parameters and 5.48G floating point operations (FLOPs), achieving excellent mean Intersection-over-Union (mIoU). Further experiments on the publicly available LoveDA and CITY-OSM datasets have been conducted to further validate the effectiveness of the proposed basic and large model, and outstanding mIoU results have been achieved. All codes are available on https://github.com/GtLinyer/LOANet.

# INTRODUCTION

Uncrewed aerial vehicles (UAVs) have some advantages such as being less susceptible to atmospheric interference, low flight altitude, high resolution and low operating costs (*Osco et al., 2021*). They have been widely used in land surveying and mapping, ecological environment monitoring, resource survey and classification, *etc*. Compared with other aerial photography collection methods, the high-resolution remote sensing images taken by UAVs are more suitable for extracting various important ground objects such as roads and houses.

In recent years, with the rapid development of deep learning, models based on convolutional neural networks have shown superior performance in some computer vision tasks such as image classification, detection, and segmentation. Compared with traditional machine learning algorithms with manual feature extraction, deep learning can automatically extract features including color, texture, shape and spatial position relationship of the image. Fully convolutional network (FCN) (*Long, Shelhamer & Darrell, 2015*), as the first semantic segmentation of natural images only using convolutional operation, realized the pixel-level classification of the images. Then, there are more semantic segmentation models (*Badrinarayanan, Kendall & Cipolla, 2017*; *Ronneberger, Fischer & Brox, 2015*; *Zhao et al., 2017*; *Chen et al., 2014*; *Chen et al., 2017a*; *Chen et al., 2017b*; *Chen et al., 2018*) that significantly improved the segmentation performance. All of these algorithms predict the pixel-level labels based on the semantic information represented by image pixels. Recently, some new deep learning methods have been proposed for extracting buildings or roads from UAV remote sensing images. *Liu et al. (2019)* introduced a chain-based U-Net network to address the problem of incomplete building boundary extraction. *Boonpook, Tan & Xu (2021)* proposed a multi-feature semantic segmentation network for extracting buildings from UAV photogrammetry. Additionally, *Sultonov et al. (2022)* designed a lightweight hybrid method based on U-Net for road extraction. *Li et al. (2019)* achieved good results in road extraction from UAV images by improving D-LinkNet to obtain BD-LinkNetPlus. However, these methods can only extract buildings or roads separately. Therefore, a lightweight network is needed that can simultaneously extract buildings and roads from UAV images.

Different from natural images, UAV remote sensing images are often very large in size, usually with a resolution of tens of thousands by tens of thousands. UAV remote sensing image segmentation is a challenging task due to the large variations in the size, shape, color, and location of the ground objects, as shown in Fig. 1. Figures 1A and 1D show different styles of buildings, even though they belong to the same ground object. In other cases, the surface feature elements of different ground objects may be similar. As shown in Figs. 1B and 1E, the top of a concrete building is very similar to the surface of a concrete road, leading to difficulties in feature extraction. There is a lot of greenery on both sides or in the center of the road, and vehicles on the road cover the road, which would affect the road surface segmentation results, as shown in Figs. 1C and 1F. Additionally, buildings with different scales are likely to affect the performance of the model, requiring a model with a strong ability to extract multi-scale context features.

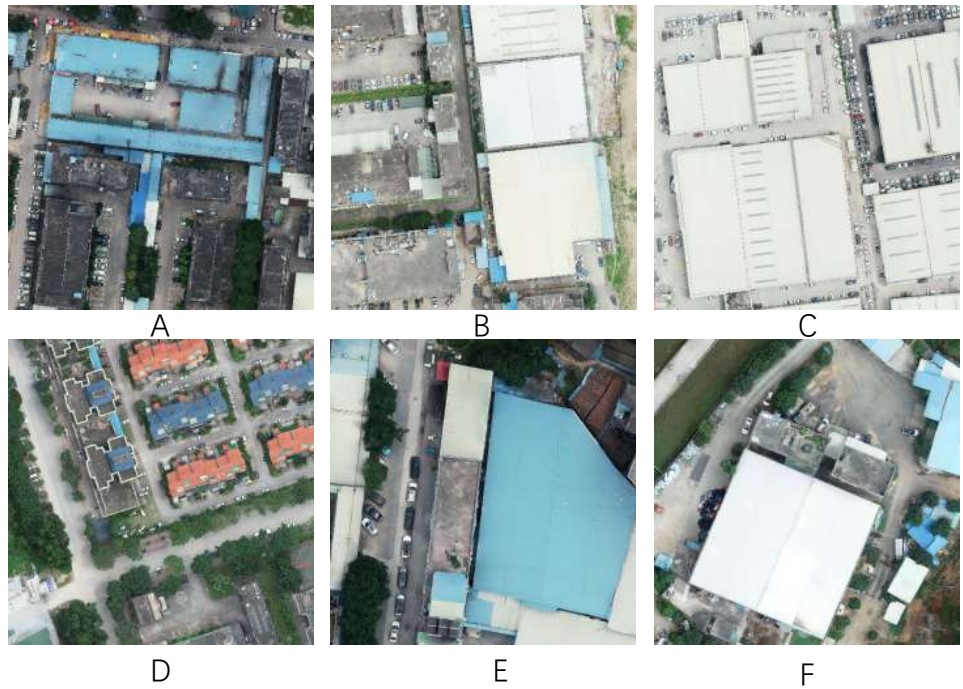

**Figure 1 Some examples from our private dataset.** Image source credit: Xiaoxiang Han. (2023). A dataset of aerial images taken by UAV that we collected [Data set]. Zenodo. https://doi.org/10.5281/zenodo.7809659. CC-BY 4.0. https://creativecommons.org/licenses/by/4.0/legalcode.

To address these issues, we propose a new deep neural network structure called LOANet for the extraction of buildings and roads from UAV remote sensing images. While DenseNet (*Huang et al., 2017*) has strong feature extraction ability, its parameters and calculations are too large. Moreover, *Liu et al. (2022)* developed a convolutional neural network (CNN) called ConvNeXt, which achieved superior performance beyond Swin-Transformer (*Liu et al., 2021*) by improving ResNet (*He et al., 2016*). Inspired by this, we propose a lightweight, high-performance convolutional neural network named LDCNet, which serves as the backbone structure of LOANet for semantic segmentation of remote sensing images. Our goal is to extend LDCNet to become a new generation backbone network for lightweight semantic segmentation algorithms. LOANet is an encoder–decoder structure that employs LDCNet as the encoder, which has demonstrated excellent segmentation results. In the decoder, we propose an object attention module (OAM) that is combined with the spatial pyramid pooling (ASPP) (*Chen et al., 2017b*). The feature pyramid network (FPN) (*Lin et al., 2017a*) module is used to fuse multi-scale features after the ASPP and before the OAM. These related works can be viewed more clearly in Table 1.

The main contributions of this article are as follows.

1. A new network based on an encoder–decoder structure for the extraction of feature elements from UAV remote sensing images is proposed. The proposed network, called LDCNet, is employed as a lightweight encoder in this article to reduce the model's parameters and accelerate the computational speed. An object attention module is

**Table 1  Related works in semantic segmentation of remote sensing land cover features.**

| Method | Dataset | Classes | mIoU (%) |
|---|---|---|---|
| Chain FCN (*Liu et al., 2019*) | Yizheng, Jiangsu, China | Buildings | 96.14 |
| Multi-Feature Semantic Segmentation (*Boonpook, Tan & Xu, 2021*) | Chongqing and Wuhan, China | Buildings | 88.97 |
| Mixer U-Net (*Sultonov et al., 2022*) | Massachusetts Roads dataset (*Mnih, 2013*) | Roads | 80.75 |
| BD-LinkNetPlus (*Li et al., 2019*) | Massachusetts Roads dataset (*Mnih, 2013*) | Roads | 59.45 |

proposed as an effective decoder to fuse and refine the target objects and boundary features.

2. A lightweight and high-performance backbone network called LDCNet is developed, incorporating design ideas from ConvNeXt and DenseNet. LDCNet serves as an efficient lightweight network for feature extraction from UAV remote sensing images, achieving this with fewer parameters.

3. Extensive experiments are carried out to verify the effectiveness and feasibility of the proposed method on our private database and two public datasets, namely LoveDA and CITY-OSM. Several popular backbone networks and semantic segmentation algorithms are used for comparison with the proposed method in the semantic segmentation of UAV remote sensing images. The experimental results show that the proposed method can effectively extract roads and buildings with higher accuracy compared to the other networks used for comparison.

# RELATED WORKS

## Semantic segmentation in remote sensing

Fully convolutional networks (FCN), as the first semantic segmentation network based on deep learning, can accept the input size of any size. To segment the remote sensing images with more complex, there are some improved FCN-based networks proposed to enhance the segmentation performance. *Maggiori et al. (2016)* designed a dual-scale neuron module based on FCN for semantic segmentation of remote sensing images, which balances the accuracy of recognition and localization. *Liu et al. (2017)* proposed an improved FCN to high-resolution remote sensing image segmentation. However, FCN has limited ability to extract objects of very small size or very large size (*Long, Shelhamer & Darrell, 2015*). *Fu et al. (2017)* adopted dilated Atrous convolution to optimize the FCN model and used conditional random field (CRF) to post-process preliminary segmentation results, which result in a significant improvement over previous networks. Atrous convolution can increase the receptive field of the convolution while maintaining the spatial resolution of the feature map.

U-Net (*Ronneberger, Fischer & Brox, 2015*) is another popular semantic segmentation network, which was first used in medical image segmentation. *Li et al. (2018)* proposed a network to segment the land and sea of high-resolution remote sensing images based on U-Net. *Cheng et al. (2020)* developed a network called HDCUNet combining U-Net with

Hybrid Dilated Convolution (HDC) for fast and accurate extraction of coastal farming areas. It avoids meshing and increases the receptive field. *Wang et al. (2022d)* designed a U-Net with two decoders and introduced spatial attention and channel attention.

In addition, some researches (*Badrinarayanan, Kendall & Cipolla, 2017*; *Song et al., 2020*) improved SegNet (*Badrinarayanan, Kendall & Cipolla, 2017*) for semantic segmentation of remote sensing images. *Weng et al. (2020)* proposed an SR-SegNet using a separable residual algorithm for water extraction from remote sensing images. *Kniaz (2019)* developed a network called GeoGAN based on the Generative Adversarial Network(GAN) (*Goodfellow et al., 2020*) to extract waters in different seasons. Recently, some transformer-based or the combination of transformer and CNN models (*Wang et al., 2022c*; *Li et al., 2021a*; *Zhang et al., 2022*) promoted the semantic segmentation performance with the transformer' advantage of global receptive field. However, the large parameters of transformer affects the calculation speed of these proposed models.

## Lightweight network

In order to deploy the network models on devices with limited resources, it is necessary to design lightweight and efficient networks. The MobileNet series (*Howard et al., 2017*; *Sandler et al., 2018*; *Howard et al., 2019*) is a classic lightweight network that applies depthwise separable convolutions (*Howard et al., 2017*), inverted residuals (*Sandler et al., 2018*), linear bottlenecks (*Sandler et al., 2018*), and lightweight channel attention modules (*Howard et al., 2019*). It can achieve higher accuracy with fewer parameters and lower calculation costs. SqueezeNet (*Iandola et al., 2016*), another lightweight network, used the Fire module for parameter compression. EffectionNet (*Tan & Le, 2019*) balanced the three dimensions of depth, width, and resolution well, and scales these three dimensions uniformly through a set of fixed scaling factors. GhostNet (*Han et al., 2020*) obtains redundant information by designing cost-efficient, which ensures the model can fully understand the input data. MicroNet li2021Textmu met used Micro-Factorized convolution and Dynamic Shift-Max to reduce the amount of calculation and improve network performance.

## Context feature extraction

By fully considering the context information can significantly boost the semantic segmentation performance and solve the problem of receptive field scale. To aggregate multi-scale contextual features, PSPNet (*Zhao et al., 2017*) used a pyramid pooling module, which connects four global pooling layers of different sizes in parallel, pools the original feature maps to generate feature maps of different levels, and restores them to the original size after convolution and upsampling. DeepLabV3 (*Chen et al., 2017b*) introduced the atrous spatial pyramid pooling(ASPP) module, which utilized different hole rates to construct convolution kernels of different receptive fields to obtain multi-scale context information. GCNet (*Cao et al., 2019*) employed a self-attention mechanism to obtain global context information. GCN (*Peng et al., 2017*) enlarged the receptive field by increasing the size of the convolution kernel. The convolution kernel up to 15 ×15 in size in GCN proved that the large convolution kernel has a stronger ability to extract context features.

## PROPOSED METHOD

### Overall structure of LOANet

The proposed lightweight and efficient semantic segmentation network, referred to as LOANet, is shown in Fig. 2. The proposed LOANet is an encoder–decoder structure. In the encoder part, LOANet takes our proposed LDCNet as a backbone, which can reduce model parameters and accelerate the computational speed. First, an image of size 512 ×512 is input into the backbone network LDCNet. And after feature extraction operation by LDCNet, four different levels of feature maps are output. The first three feature maps are respectively input into the Atrous spatial pyramid pooling module (ASPP) (_Chen et al., 2017b_) to extract multi-scale context information. The three outputs are fed into the corresponding 1 ×1 convolution operation in the feature pyramid network (FPN) (_Lin et al., 2017a_) and upsampled to the same size. Four feature maps of the same size are then concatenated as the input of OAM proposed by us. After the feature map is processed by OAM, it will produce a rough segmentation result and a feature map with object context information. This rough segmentation result is taken as one of the outputs of the network, and the feature maps are fed into the next step for further processing. Then, the feature map is processed by a refinement classification head to refine the segmentation edges. It consists of a 3 ×3 depthwise separable convolution and a 1 ×1 convolution. Finally, the feature map is upsampled to the size of the input image.

### LDCNet

Inspired by ConvNeXt and DenseNet, this article develops a backbone network, called LDCNet. The structure and densely connected blocks are shown in Fig. 3. ConvNeXt improved the classical ResNet by introducing some of the latest ideas and technologies of Transformer network to enhance the performance of CNN. Macro design, reference of ResNeXt's design ideas (_Xie et al., 2017_), inverted bottleneck layer, large convolution kernel and micro design of various layers are the five main optimization design of ConvNeXt. In the macro design of ConvNeXt, the stacking ratio of multi-stage blocks is 1:1:3:1. The number of blocks stacked in the third stage is larger. This improves the model accuracy of ConvNeXt. Following the design, we set the stacking ratio of blocks in each stage of LDCNet to 1:1:3:1. The specific layers of each stage are 2, 2, 6 and 2 respectively.

ConvNeXt designs the effective inverted bottlenecks block with a 3 ×3 depthwise separable convolution, shown in Figs. 4A and 4B. This structure can partially reduce the parameter scale of the model while slightly improving the accuracy rate. Considering the multi-scale feature extraction ability of Inception block in GoogLeNet (_Szegedy et al., 2015_), a new bottleneck layer with two branches is proposed in this article. One of the branches is a depthwise separable convolution with a 7 ×7 convolution kernel, and the other is a depthwise separable convolution with a 3 ×3 convolution kernel. After adding the output feature maps of two branches, and then concatenating with the input feature map, the output feature map of the proposed bottleneck layer is produced, as shown in Fig. 4C. In recent years, some new studies (_Peng et al., 2017_; _Ding et al., 2022_; _Guo et al., 2022_) have stated the large convolution kernels are more efficient for enlarging the receptive field. In

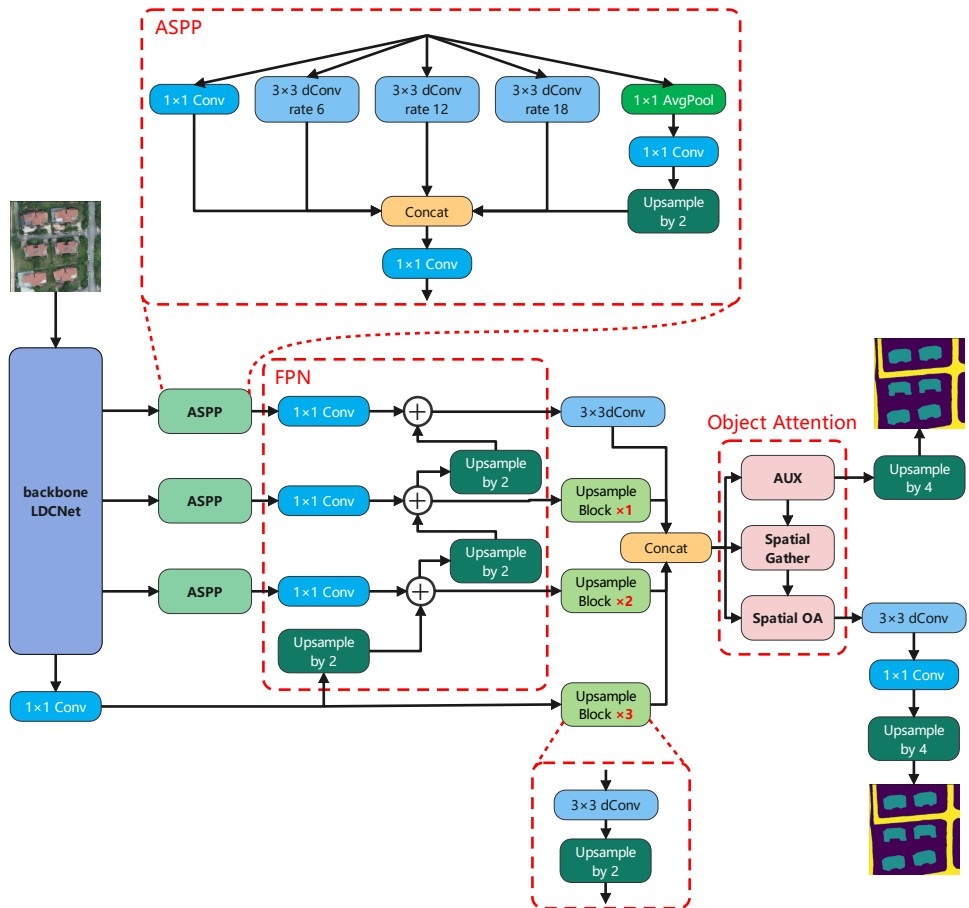

**Figure 2** **The overall architecture of LOANet, and the details of ASPP.** Image source credit: Xiaoxiang Han. (2023). A dataset of aerial images taken by UAV that we collected [Data set]. Zenodo. https://doi.org/10.5281/zenodo.7809659. CC-BY 4.0. https://creativecommons.org/licenses/by/4.0/legalcode.

order to obtain a higher computational efficiency, a 7 ×7 convolution kernel is used at the beginning of one branch in the proposed bottleneck layer.

As we all know, batch normalization (BN) (*Ioffe, 2017*) is most popular optimization process in computer vision tasks by computing the mean and variance of each minibatch and pulls it back to a standard normal distribution with mean 0 and variance 1 to make neural network training faster and stabler However, BN also has may some drawbacks detrimental to the model performance (*Wu & Johnson, 2021*). Transformers use a simpler layer normalization (LN) (*Ba, Kiros & Hinton, 2016*), which is more common in natural language processing tasks. ConvNeXt replaces all BNs with LNs to improve the model performance. Since the features of our remote sensing images depend on the statistical parameters between different samples. it is inappropriate to replaces all BNs with LNs in the task of extracting buildings and roads from UAV remote sensing images. Therefore, we use an LN after a 3 ×3 depthwise separable convolution in the bottleneck layer, and replaced the BN in the transition layer with LN.

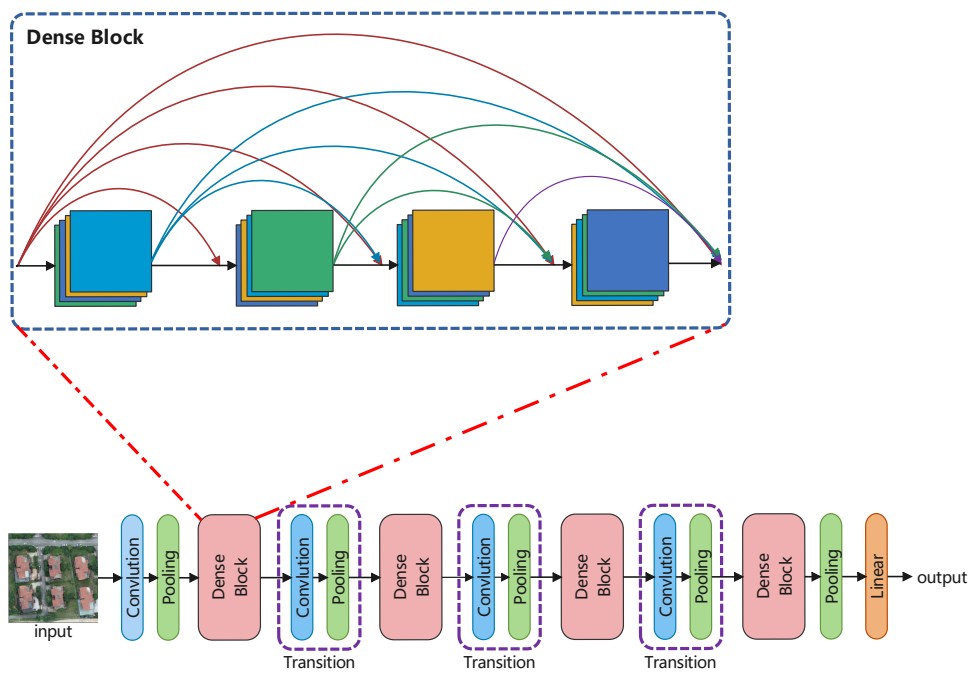

**Figure 3** **The overall structure and densely connected blocks of LDCNet.** Image source credit: Xiaoxiang Han. (2023). A dataset of aerial images taken by UAV that we collected [Data set]. Zenodo. https://doi.org/10.5281/zenodo.7809659. CC-BY 4.0. https://creativecommons.org/licenses/by/4.0/legalcode.

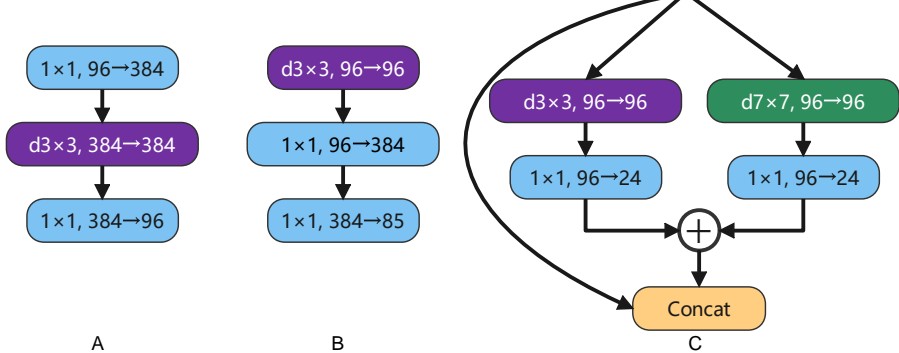

**Figure 4** **A is the inverted bottleneck layer designed by the authors of ConvNeXt. B is the actual bottleneck layer of ConvNeXt. C is our bottleneck layer for LDCNet.**

## Object attention module

We proposed an object attention module (OAM). OAM considers the relationship between a pixel and its context pixels, and aggregates similar context pixel representations with higher weights. Unlike regular attention mechanisms, OAM constructs attention into object regions and exploits the relationship between pixels and object regions.

OAM is different from global context feature extraction methods or global attention mechanisms, as it computes the similarity between each pixel and each object region using

dot-product attention mechanism, and fuses object region features to refine pixel features. This produces multiple weights for each pixel, indicating how much it belongs to each object region. Then, OAM uses these weights to compute a weighted sum of object region features, and concatenates it with the original pixel features.

The OAM consists of three steps: soft object region extraction, object region computation, and object attention computation for each position. It is mainly based on the scaled dot-product self-attention from the Transformer structure. The input to attention consists of: a set of $A_q$ queries $Q \in \mathbb{R}^{d \times A_q}$, a set of $A_{kv}$ keys $K \in \mathbb{R}^{d \times A_{kv}}$, and a set of $A_{kv}$ values $V \in \mathbb{R}^{d \times A_{kv}}$. The attention weight $a_j$ is computed as the Softmax normalization of the dot product between query $q_i$ and key $k_j$:

$$B_i = \sum_{j=1}^{A_{kv}} e^{\frac{1}{\sqrt{d}} q_i^\top k_j} \tag{1}$$

$$m_{ij} = \frac{e^{\frac{1}{\sqrt{d}} q_i^\top k_j}}{Z_i} \tag{2}$$

The attention output for each query $q_i$ is the aggregation of values weighted by attention weights:

$$Attention(q_i, K, V) = \sum_{j=1}^{A_{kv}} m_{ij} Vj \tag{3}$$

For object attention, the calculation formula of the relationship between each pixel and each object area is as follows:

$$y_{ik} = \frac{e^{F(x_i, f_k)}}{\sum_{j=1}^{K} e^{F(x_i, f_j)}}. \tag{4}$$

Among them, $F(x, f) = u(x)^\top v(f)$ is the denormalized relationship function, $u(\cdot)$ and $v(\cdot)$ are two transformation functions implemented by $1 \times 1\text{Conv} \rightarrow \text{BN} \rightarrow \text{ReLU}$.

## Atrous Space Pyramid Pooling (ASPP)

In order to obtain a large receptive field without losing spatial resolution and increasing computation, ASPP uses multiple parallel dilated convolutional layers with different sampling rates. The features which are extracted for each sampling rate are further processed in separate branches and fused to generate the final result.

Atrous convolution can expand the receptive field of the convolution kernel without loss of resolution. Using ASPP can achieve multi-scale feature extraction through different receptive fields and upsampling. In a two-dimensional convolution, for each position i on the feature y of the convolution output and the corresponding convolution kernel w, for the input x, the calculation of the dilated convolution is as follows:

$$y[i] = \sum_k x[i + r \cdot k] w[k] \tag{5}$$

where r is the hole rate, which represents the sampling step size of the convolution kernel on the input x of the convolution operation. k represents the position of the convolution kernel parameter. If the convolution kernel size is 3, then $k = 0,1,2$. The receptiv

## EXPERIMENT

### Dataset

A private dataset of UAV-borne remote sensing images with a resolution between 10000 ×10000 and 20000 ×20000 is constructed. Each remote sensing image which corresponds to the red (R), green (G), and blue (B) bands is cropped to the same size of 512 ×512. All the images are divided into a training set of 2431 images, a validation set of 945 images, and a test set of 676 images. The category labels of two important ground objects, *i.e.,* road and building, are manually annotated. The training and validation sets are from a city in Guangdong, China, and the test set is from a place along the Yangtze River, China.

Furthermore, in order to verify the competitive performance and model generalization ability of our proposed model in the task of semantic segmentation of aerial remote sensing images, we test it on two public datasets, *i.e.,* LoveDA (*Wang et al., 2021*) and CITY-OSM (*Kaiser et al., 2017*). LoveDA dataset includes the cities of Nanjing, Changzhou and Wuhan in China, with a total of 5,987 high spatial resolution (0.3 m) remote sensing images, with a training set of 2,522 images, a validation set of 1669 images, and a test set of 1796 images. The pixels in each image are divided into six categories, namely road, building, water, barren, forest, agricultural, and background. For the sake of fairness, we only take two types of ground objects, building and road. CITY-OSM dataset includes Berlin and Potsdam in Germany, Chicago in the United States, Paris in France, Zurich in Switzerland and Tokyo in Japan, with a total of 1,641 aerial images. Its labels are two types of ground objects, *i.e.,* building and road, which are consistent with the label categories of our private dataset. We remove the images that are obviously mislabeled in this dataset, for example the entire image is labelled as a building or a road. All the images are cropped and scaled to the size of 512 ×512, which are divided into a training set of 10621 images, a validation set and a test set of 3401 images.

### Training details

Our model was developed utilizing Python3.8 and the PyTorch1.12.1 machine learning framework. PyTorch-Lightning1.6.5, which builds upon PyTorch, was utilized to further expedite the process. For comparison and ablation experiments, we employed Torchvision0.13.1′s backbone network.

To train the model, a GPU server boasting an Intel Core i9-10900X CPU, two Nvidia RTX3080 GPUs (10GB), 32GB RAM was utilized.

The batch size of data was altered based on the network to guarantee maximum memory utilization. Additionally, the data reading program was equipped with 16 threads. The initial learning rate was set at 1e−3, dynamic learning rate adjustment was facilitated *via* ReduceLROnPlateau, and optimization fell under the responsibility of AdamW (*Loshchilov & Hutter, 2018*). Automatic mixed precision was employed during training, with a loss function of FocalLoss (*Lin et al., 2017b*). This function aimed to reduce the weight of

easily-classified samples while prioritizing harder-to-classify samples. Training was carried out over a period of 100 epochs. Its formula is as follows:

$$FL(p_t) = -\alpha_t(1-p_t)^\gamma \log(p_t) \tag{6}$$

p $\in [0,1]$ is the model's estimated probability of the labeled class, $\gamma$ is an adjustable focusing parameter, and $\alpha$ is a balancing parameter. We set $\gamma$ to 2 and $\alpha$ to 0.25.

## Evaluation metrics

In order to evaluate the performance and efficiency of the proposed LOANet model, our evaluation indicators are divided into two categories. The first category is to evaluate the accuracy of the network, including overall accuracy (OA), average F1-score (F1) and mean intersection over union (mIoU). The results of these evaluation indicators are calculated based on the confusion matrix, where TP indicates the number of true positive categories, TN indicates the number of true negative categories, FP indicates the number of false positive categories, and FN indicates the number of false negative categories.

Overall accuracy (OA) is used to measure the overall accuracy of the model prediction results:

$$OA = \frac{TP + TN}{TP + TN + FP + FN} \tag{7}$$

The F1-score (F1) indicates the comprehensive consideration of precision and recall:

$$Precision = \frac{TP}{TP + FP} \tag{8}$$

$$Recall = \frac{TP}{TP + FN} \tag{9}$$

$$F1 = 2\frac{Precision \times Recall}{Precision + Recall} = \frac{2TP}{2TP + FP + FN} \tag{10}$$

Intersection over union (IoU) is used to measure the ratio of the intersection and union of the predicted results of a certain category and the true value of the model:

$$IoU = \frac{TP}{TP + FN + FP}. \tag{11}$$

The second category is to evaluate the scale of the network, including floating point operations (FLOPs) for evaluating complexity, frames per second (FPS) for evaluating speed, memory usage (MB) and the number of model parameters (M) to evaluate memory requirements.

## Experimental results on the private dataset

In this section, LOANet is compared with some other semantic segmentation models on the constructed private dataset. These models include GCNet (*Cao et al., 2019*), PSPNet *Zhao et al., 2017*), SegFormer *Xie et al., 2021*), and A2FPN *Li et al., 2022*), DC-Swin *Wang et al., 2022b*), BuildFormer (*Wang et al., 2022a*) proposed for remote sensing semantic

**Table 2  Quantitative results on the private dataset.**

| Method | Backbone | Background (%) | building (%) | road (%) | OA (%) | mean F1 (%) | mIoU (%) |
|---|---|---|---|---|---|---|---|
| DC-Swin | – | 76.47 | 55.74 | 23.17 | 87.08 | 65.44 | 52.14 |
| SegFormer | – | 78.55 | 56.94 | 36.23 | 88.37 | 71.24 | 57.24 |
| GCNet | ResNet18 | 79.55 | 61.53 | 37.72 | 89.09 | 73.19 | 59.62 |
| A2FPN | ResNet18 | 81.46 | 66.01 | 41.73 | 90.33 | 76.06 | 63.26 |
| PSPNet | ResNet18 | 82.56 | 67.52 | 43.87 | 90.89 | 76.92 | 64.22 |
| BuildFormer | – | 83.26 | 68.56 | 47.76 | 91.34 | 78.95 | 66.53 |
| LOANet (ours) | ResNet18 | 83.01 | 67.59 | 43.63 | 91.13 | 77.38 | 64.83 |
| LOANet (ours) | LDCNet (ours) | **85.73** | **75.01** | _52.70_ | **92.83** | **82.35** | **71.12** |
| LOANet-Large (ours) | LDCNet-Large (ours) | _85.72_ | _74.96_ | **52.71** | _92.79_ | _82.29_ | _71.11_ |

**Notes.**
Bold values represent the best performance indicators. Underlined values represent the second best performance indicators.

segmentation. The backbone network of the other compared models is ResNet18, and LOANet uses the proposed backbone network LDCNet. LOANet is divided into two versions, the basic version and the large version. The only difference is that the depth of LDCNet is different. The number of dense block stacks in the basic version is 2, 2, 6, 2, while the number of stacks in the large version is 6, 6, 18, 6. From Table 2, it is clear seen that the proposed model outperforms other models in term of the numerical results. The mIoU of the proposed model is 4.59% higher than that of the powerful BuildFormer. The IoU of the background, buildings and roads increased by 2.47%, 6.45% and 4.04% respectively compared with BuildFormer. Furthermore, the overall accuracy and mean F1-score of the proposed model are 0.94% and 3.40% higher than the powerful BuildFormer. However, the large-scale version of LOANet does not perform as well as the base version on our smaller-scale dataset, and it overfits. Moreover, LOANet using the proposed LDCNet as the backbone outperforms LOANet with ReseNet18 as the backbone. A comparison of the visual effects *via* different methods is depicted in Fig. 5. It can be seen from the figure that the proposed model handles edges better than other methods. And the extracted buildings did not generate a large number of voids. It can extract relatively small roads.

## Experimental results on the public dataset LoveDA

In this section, the compared experiments are conducted on LoveDA dataset. The compared models are the same as subsection. The comparison results are shown in Table 3 and Fig. 6. The proposed model outperforms the other compared models in both quantitative and qualitative results on the LoveDA dataset. The mIoU of LOANet is 1.22% higher than the powerful BuildFormer. The IoU of the background and buildinghouses increased by 0.71% and 4.04% respectively compared with BuildFormer. The IoU of the road is slightly lower than that of BuildFormer. Furthermore, the overall accuracy and mean F1-score of the proposed model are 0.45% and 0.97% higher than the powerful BuildFormer. The performance of the large version of LOANet is slightly lower than that of the basic version. The performance of LDCNet as a backbone network surpasses ResNet18. It can be seen from the Fig. 6. That the proposed model extracts the whole building house area much

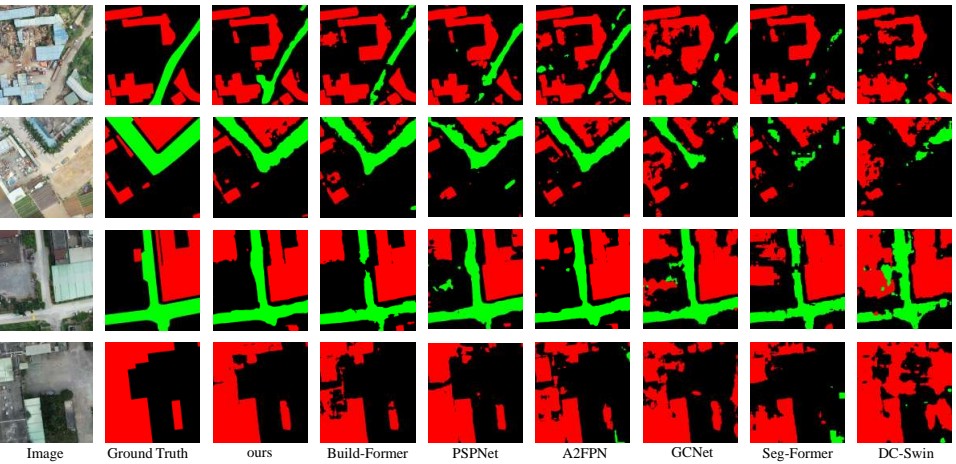

**Figure 5** **Qualitative results on the private dataset.** Image source credit: Xiaoxiang Han. (2023). A dataset of aerial images taken by UAV that we collected [Data set]. Zenodo. https://doi.org/10.5281/zenodo.7809659. CC-BY 4.0. https://creativecommons.org/licenses/by/4.0/legalcode.

**Table 3** **Quantitative results on the public dataset LoveDA.**

| Method | Backbone | Background (%) | Building (%) | Road (%) | OA (%) | Mean F1 (%) | mIoU (%) |
|---|---|---|---|---|---|---|---|
| DC-Swin | – | 88.41 | 32.94 | 24.34 | 92.47 | 60.85 | 48.57 |
| SegFormer | – | 90.85 | 36.00 | 33.89 | 93.60 | 66.10 | 53.31 |
| GCNet | ResNet18 | 89.83 | 37.02 | 35.04 | 93.56 | 66.85 | 53.90 |
| A2FPN | ResNet18 | 90.18 | 38.82 | 43.59 | 93.82 | 70.49 | 57.59 |
| PSPNet | ResNet18 | 91.07 | 48.55 | 44.05 | 94.81 | 74.05 | 61.43 |
| BuildFormer | – | 91.94 | 49.97 | **50.26** | 94.97 | 76.44 | 64.05 |
| LOANet (ours) | ResNet18 | 91.47 | 44.98 | 47.49 | 94.66 | 74.00 | 61.35 |
| LOANet (ours) | LDCNet (ours) | **92.65** | **54.01** | 49.13 | **95.42** | **77.41** | **65.27** |
| LOANet-Large (ours) | LDCNet-Large (ours) | 92.59 | 53.96 | 50.02 | 95.39 | 77.00 | 65.21 |

**Notes.**
Bold values represent the best performance indicators. Underlined values represent the second best performance indicators.

better than other methods. And it extracts small roads better than most other methods. The proposed model has good detail extraction ability.

## Experimental results on the public dataset CITY-OSM

In this section, a large-scale data set named CITY-OSM is used to demonstrate the performance of the proposed LOANet. From Table 4, On such our basic model performs slightly worse than the two models of PSPNet and BuildFormer. This demonstrates the good generalization ability of our basic model on smaller datasets. However, the large version of LOANet outperforms other methods. The mIoU is 0.28% higher than that of the powerful BuildFormer, and the IoU of buildinghouses and roads are 0.21% and 0.51% higher, respectively. It is obvious from Fig. 7. that the proposed model has strong road extraction ability.

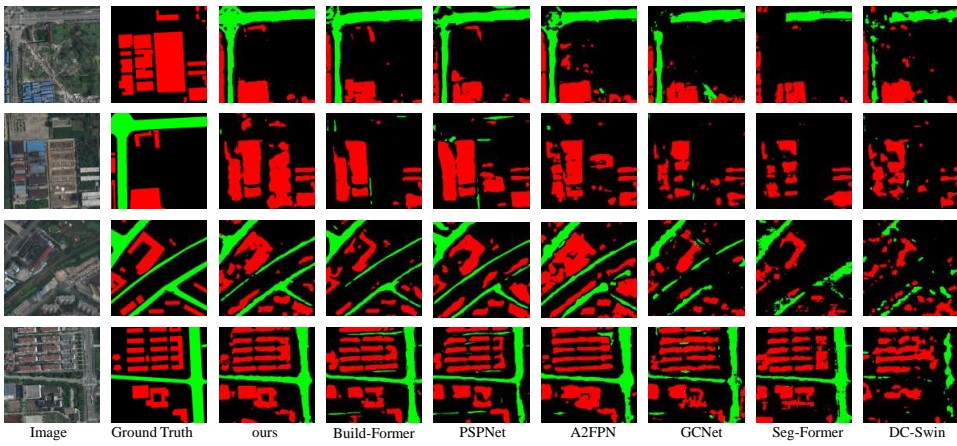

| Image | Ground Truth | ours | Build-Former | PSPNet | A2FPN | GCNet | Seg-Former | DC-Swin |

**Figure 6** **Qualitative results on the public dataset LoveDA.** Image source credit: Junjue, Wang, Zhuo, Zheng, Ailong, Ma, Xiaoyan, Lu, & Yanfei, Zhong. (2021). LoveDA: A Remote Sensing Land-Cover Dataset for Domain Adaptive Semantic Segmentation [Data set]. Thirty-fifth Conference on Neural Information Processing Systems (NeurIPS 2021). Zenodo. https://doi.org/10.5281/zenodo.5706578. CC-BY 4.0. https://creativecommons.org/licenses/by/4.0/legalcode.

**Table 4** **Quantitative results on the public dataset CITY-OSM.**

| Method | Backbone | Background (%) | Building (%) | Road (%) | OA (%) | Mean F1 (%) | mIoU (%) |
|---|---|---|---|---|---|---|---|
| DC-Swin | – | 63.71 | 68.79 | 55.06 | 85.42 | 76.79 | 62.52 |
| SegFormer | – | 67.35 | 72.69 | 61.73 | 87.48 | 80.34 | 67.26 |
| GCNet | ResNet18 | 66.67 | 71.76 | 63.44 | 87.32 | 80.39 | 67.28 |
| A2FPN | ResNet18 | 71.12 | 75.72 | 77.65 | 89.26 | 83.34 | 71.50 |
| PSPNet | ResNet18 | 75.40 | 79.01 | 73.10 | 91.06 | 86.24 | 76.04 |
| BuildFormer | – | 75.36 | 79.30 | 72.79 | 91.08 | 86.22 | 75.82 |
| LOANet (ours) | ResNet18 | 73.17 | 76.54 | 70.76 | 90.04 | 84.70 | 73.49 |
| LOANet (ours) | LDCNet (ours) | 73.95 | 78.25 | 70.95 | 90.49 | 85.28 | 74.39 |
| LOANet-Large (ours) | LDCNet-large (ours) | **75.51** | **79.51** | **73.30** | **91.17** | **86.41** | **76.10** |

**Notes.**
Bold values represent the best performance indicators.

## Evaluation of model efficiency

It can be seen from the Table 5 that the proposed model, whether it is the basic version or the enlarged version, has smaller parameters and model sizes than other models. In addition, the FLOPs of the proposed basic version of the model is also the smallest. Therefore, the proposed model can be applied in a variety of hardware performance-limited scenarios. However, the proposed large model has shortcomings; specifically, its FPS is not the highest, which is an aspect we aim to improve in the next step.

## Ablation study

To evaluate the contribution of each component of the proposed LOANet the ablation experiments are conducted using LoveDA dataset and CITY_OSM dataset, as shown in

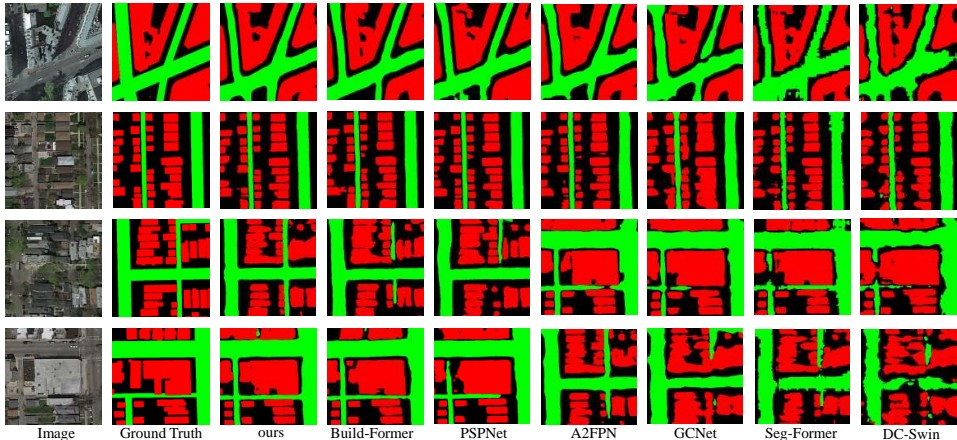

| Image | Ground Truth | ours | Build-Former | PSPNet | A2FPN | GCNet | Seg-Former | DC-Swin |

**Figure 7** **Qualitative results on the public dataset CITY_OSM.** Image source credit: Kaiser Pascal, Wegner Jan Dirk, Lucchi Aurelien, Jaggi Martin, Hofmann Thomas, & Schindler Konrad. (2017). Learning Aerial Image Segmentation From Online Maps [Data set]. In IEEE Transactions on Geoscience and Remote Sensing (Vol. 55, Number 11, pp. 6054–6068). https://doi.org/10.1109/TGRS.2017.2719738. CC-BY 4.0. https://creativecommons.org/licenses/by/4.0/legalcode.

**Table 5** **Evaluation of model efficiency.**

| Method | Params (M) | Size (MB) | FLOPs (G) | FPS |
|---|---|---|---|---|
| DC-Swin | 118 | 237.858 | 126.15 | 34.7 |
| SegFormer | 7.7 | 15.436 | 13.11 | 115.2 |
| GCNet | 61.5 | 122.933 | 9.84 | 182.6 |
| A2FPN | 12.2 | 24.318 | 13.21 | 212.4 |
| PSPNet | 24.3 | 48.648 | 96.53 | 98.5 |
| BuildFormer | 40.5 | 81.038 | 116.22 | 52.4 |
| LOANet (ours) | **1.4** | **2.628** | **5.48** | **212.6** |
| LOANet-large (ours) | _6.1_ | _12.271_ | 13.69 | 108.4 |

**Notes.**
Bold values represent the best performance indicators. Underlined values represent the second best performance indicators.

Table 6. After adding the ASPP module to the baseline model, mIoU on the two public datasets increased by 0.55% and 0.42%, respectively. After adding the OAM to the baseline model, mIoU on the two public datasets increased by 1.55% and 0.88%, respectively. After the proposed model aggregates these two modules, mIoU on two public datasets improves by 2.03% and 0.69% compared to the baseline model, respectively.

In addition, we also studied the impact of using different backbone networks on the performance of the model. As shown in Table 7, the proposed LOANet surpasses mainstream backbone networks. Compared to ResNet18, the proposed method showed an improvement of 3.29% and 0.9% in mIoU on two common datasets, respectively.

**Table 6  Ablation study.**

| Dataset | Method | Overall accuracy | Mean F1-Score | mIoU |
|---------|--------|------------------|---------------|------|
| LoveDA | Baseline | 94.79 | 75.77 | 63.24 |
| | Baseline + ASPP | 95.21 | 76.14 | 63.79 |
| | Baseline + OAM | 95.29 | 77.02 | 64.79 |
| | Baseline + ASPP + OAM | **95.42** | **77.41** | **65.27** |
| CITY_OSM | Baseline | 90.19 | 84.82 | 73.70 |
| | Baseline + ASPP | 90.36 | 85.11 | 74.12 |
| | Baseline + OAM | 90.32 | 85.03 | 74.00 |
| | Baseline + ASPP + OAM | **90.49** | **85.28** | **74.39** |

**Notes.**
Bold values represent the best performance indicators.

**Table 7  Comparison of different backbones.**

| Dataset | Backbone | Overall accuracy | Mean F1-Score | mIoU |
|---------|----------|------------------|---------------|------|
| LoveDA | ResNet18 | 94.66 | 74.02 | 61.35 |
| | DenseNet121 | 95.16 | 76.25 | 63.81 |
| | Swin Transformer | 95.08 | 75.98 | 63.55 |
| | MobileNetV3 | 94.78 | 74.22 | 61.62 |
| | LDCNet | **95.42** | **77.41** | **65.27** |
| CITY_OSM | ResNet18 | 90.04 | 84.70 | 73.49 |
| | DenseNet121 | 90.38 | 85.14 | 74.32 |
| | Swin Transformer | 90.29 | 85.03 | 74.19 |
| | MobileNetV3 | 89.91 | 84.37 | 73.01 |
| | LDCNet | **90.49** | **85.28** | **74.39** |

**Notes.**
Bold values represent the best performance indicators.

## DISCUSSION

LOANet has achieved excellent performance with a smaller number of parameters and lower computational cost. Compared to larger models in the past, it is suitable for running on a wider range of hardware conditions. There are three main reasons for the outstanding performance of the proposed model. Firstly, this article customizes a lightweight and efficient backbone network LDCNet for LOANet, which extensively uses depthwise separable convolutions and some design techniques. This makes LDCNet both computationally efficient and powerful in feature extraction. Secondly, we proposed the OAM, which has more advantages in extracting details and edges. The ASPP module can obtain features of different scales of buildings or roads in remote sensing images by using different sizes of receptive fields, while the OAM can focus more on the relationships between pixels within a single category of buildings or roads. Therefore, these two modules complement each other. In addition, FPN combines semantic features of different levels, enhancing the network's ability to extract small-scale targets.

The method proposed in this article has some limitations. Although the proposed quantitative and qualitative indicators have significant advantages over other methods,

there are still many shortcomings in terms of visual effects. For example, some complex edge segmentation is not perfect, and the segmentation of some complex background areas is not accurate enough. These are common problems in remote sensing image segmentation and are also challenging issues. To address these issues, we will conduct further research on methods to solve them in the future. In particular, for the segmentation of complex background areas, the model needs to have a more powerful integration capability of contextual information, while not overly increasing the complexity of the model to avoid limiting its wide application. In addition, we hope that the proposed method can be extended to multiple semantic segmentation domains.

## CONCLUSION

In this article, a lightweight and efficient semantic segmentation network based on encoder–decoder structure was developed for buildings and roads extraction from UAV remote sensing images. The encoder used a new lightweight and high-performance backbone network proposed in this article to accelerate model calculation and reduce model parameters. The decoder employs our proposed OAM to efficiently capture more object information. We evaluated our model on two public datasets, LoveDA and CITY-OSM, and on the private dataset. With only 1.4M parameters and 5.48G FLOPs, our basic model achieved mIoU scores of 65.27%, 74.39%, and 71.12% on these datasets, respectively, demonstrating its excellent performance.

### Funding

This work was supported by the National Natural Science Foundation of China (No. 61801288). The Excellent Doctoral/Master's Dissertation Cultivation Project of Shanghai University of Medicine & Health Sciences supported the APC for this article. The National Natural Science Foundation of China was involved in collecting and analyzing the private dataset, as well as acquiring and analyzing experimental data. The funders had no role in study design, data collection and analysis, decision to publish, or preparation of the manuscript.

### Grant Disclosures

The following grant information was disclosed by the authors:
The National Natural Science Foundation of China: 61801288.
The Excellent Doctoral/Master's Dissertation Cultivation Project of Shanghai University of Medicine & Health Sciences.
The National Natural Science Foundation of China.

### Competing Interests

The authors declare there are no competing interests.

## Author Contributions

- Xiaoxiang Han conceived and designed the experiments, performed the experiments, performed the computation work, authored or reviewed drafts of the article, and approved the final draft.
- Yiman Liu conceived and designed the experiments, performed the experiments, analyzed the data, performed the computation work, authored or reviewed drafts of the article, and approved the final draft.
- Gang Liu conceived and designed the experiments, analyzed the data, prepared figures and/or tables, and approved the final draft.
- Yuanjie Lin analyzed the data, prepared figures and/or tables, and approved the final draft.
- Qiaohong Liu analyzed the data, authored or reviewed drafts of the article, and approved the final draft.

## Data Availability

The private dataset we collected is available at Zenodo: Xiaoxiang Han. (2023). A dataset of aerial images taken by UAV that we collected [Data set]. Zenodo. Available at https://doi.org/10.5281/zenodo.7809659.

Our processed LoveDA dataset is available at Zenodo: Xiaoxiang. (2023). Our processed LoveDA dataset [Data set]. Zenodo. Available at https://doi.org/10.5281/zenodo.7809597.

Our processed CITY_OSM dataset is available at Zenodo: Xiaoxiang. (2023). Our processed CITY_OSM dataset [Data set]. Zenodo. Available at https://doi.org/10.5281/zenodo.7809595.

## Supplemental Information

Supplemental information for this article can be found online at http://dx.doi.org/10.7717/peerj-cs.1467#supplemental-information.

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
