# Peer review of "LOANet: a lightweight network using object attention for extracting buildings and roads from UAV aerial remote sensing images"

_PeerJ Computer Science, doi:10.7717/peerj-cs.1467_

## Round 0.1 · original submission · Minor Revisions

Dear authors,

Your article has a few remaining issues. We encourage you to address the concerns and criticisms of the reviewer and resubmit your article once you have updated it accordingly.

Best wishes,

Reviewer 1 ·

Basic reporting

I reviewed the work "LOANet: a lightweight network using object attention for extracting buildings and roads from UAV aerial remote sensing images" in detail. I have stated the deficiencies of the study in the articles. Researchers have proposed a new model to extract buildings and roads with deep learning from unmanned aerial vehicle (UAV) remote-sensing images. The proposed network adopts an encoder-decoder architecture in which Light Dense Connected Network (LDCNet) is developed as the encoder. In the decoder part, dual multi-scale context modules consisting of the Atrous Spatial Pyramid Pooling module (ASPP) and Object Attention Module (OAM) are designed to capture more context information from feature maps of UAV remote sensing images.

Experimental design

The values given in the summary and the values given in the results section should be reviewed. There is an ambiguity here that is difficult to understand. A table related to the subject can be added in the literature review section. Thanks to the table to be added, it will be easier to follow the studies.

Validity of the findings

In the dataset section, the number of classes, the number of data in each class, etc. Presenting information is important. Information about the LoveDA and t CITY-OSM datasets should be given. Whether the results presented in Table 6 are classification results or segmentation results should be given in detail. Limitations of the study should be discussed in the Discussion section. Information about future work should be given. The conclusion section should be expanded by using the results obtained in the study. Finally, I would like to state that the study should be re-read by the researchers in order to correct the typos.

Additional comments

I want to point out that the paper is generally well-written. I think the paper will improve if the authors address the shortcomings noted in the revision.

Reviewer 2 ·

Basic reporting

The services of language experts are required.

Experimental design

What is the innovation of the Object attention module? Please explain the innovation carefully and explain its specific function in the network.

Validity of the findings

The authors mention LDCNet as a lightweight network in the main contribution, but the FPS metric is not mentioned in the quantitative results of the two datasets. Please explain the relationship between parameters, computation, and FPS metric.

Future directions should be based on the limitation of the current study and the expansion of the work.

Annotated reviews are not available for download in order to protect the identity of reviewers who chose to remain anonymous.

---

## Round 0.2 · accepted · Accept

Dear authors,

Thank you for clearly addressing all of the reviewers' comments. Your article is accepted for publication now.

Best wishes,

Reviewer 1 ·

Basic reporting

Thank you to the researchers for making a nice revision.

Experimental design

.

Validity of the findings

.

Additional comments

.